Accepted at the ICLR 2024 Workshop on AI4Differential Equations In Science

# INVESTIGATION OF LATENT TIME-SCALES IN NEURAL ODE SURROGATE MODELS

**Ashish S. Nair**[*]
University of Notre Dame
Notre Dame, IN 46556, USA
anair@nd.edu

**Shivam Barwey**
Argonne National Laboratory
Lemont, IL 60439 USA
sbarwey@anl.gov

**Pinaki Pal**
Argonne National Laboratory
Lemont, IL 60439 USA
pal@anl.gov

**Jonathan F. MacArt**
University of Notre Dame
Notre Dame, IN 46556, USA
jmacart@nd.edu

**Romit Maulik**
Pennsylvania State University
University Park, PA 16802, USA
Argonne National Laboratory
Lemont, IL 60439 USA
rmaulik@psu.edu

## ABSTRACT

This work explores autoencoder-based neural ordinary differential equation (neural ODE) surrogate models for advection-dominated dynamical systems. Alongside predictive demonstrations, physical insight into the sources of model acceleration (i.e., how the neural ODE achieves its acceleration) is the scope of the current study. Such investigations are performed by quantifying the effect of neural ODE components on latent system time-scales using eigenvalue analysis of dynamical system Jacobians. This work uncovers the key role played by the training trajectory length on the latent system time-scales: larger trajectory lengths correlate with an increase in limiting neural ODE time-scales, and optimal neural ODEs are found to recover the largest time-scales of the full-order (ground-truth) system. Demonstration studies are performed using datasets sourced from numerical solutions of the Kuramoto-Sivashinsky equation and hydrogen-air channel detonations (compressible reacting Navier-Stokes equations).

## 1 INTRODUCTION

Since its introduction in Chen et al. (2018), the neural ODE strategy has cemented itself as a powerful scientific machine learning tool to model the evolution of dynamical systems using neural networks, and has found a vast number of applications across several fields (Finlay et al., 2020; Lee & Parish, 2020; Owoyele & Pal, 2022; Kumar et al., 2023; Poli et al., 2019). Instead of directly enforcing discrete temporal representations (e.g., as used in residual networks described in Jiang et al. (2021), or recurrent neural networks described in Reddy et al. (2019)), neural ODEs learn a continuous representation of nonlinear system dynamics using discretely sampled trajectory data. The critical advantage is that the *instantaneous* right-hand-side is modeled as a nonlinear function via a neural network, allowing the framework to leverage (a) existing time-integration schemes to execute the forecasting step, and (b) adjoint methods to enable memory-efficient backpropagation.

When equipped with autoencoder-provided latent spaces, the neural ODE conveniently outputs a functional form for the instantaneous rate-of-change of low-dimensional yet expressive latent variables, a highly useful asset for reduced-order modeling (ROM), and in turn, real-time physics simulation applications. As a result, the combined autoencoder-based neural ODE strategy has been used to develop accelerated surrogate models in a variety of scenarios (Dutta et al., 2021; Linot et al., 2023).

Of particular interest to this work is the application of neural ODE ROMs to advection-dominated systems (i.e., systems in which the governing equations are heavily influenced by advection

---

[*]Work performed during an internship at the Argonne Leadership Computing Facility.

terms) that typically describe turbulent or shock-containing fluid flows over aircrafts and in power/propulsion devices (Linot et al., 2023). If available, such ROMs could drastically accelerate simulation-augmented design loops for deploying resilient next-generation aerospace concepts (Brenner et al., 2019; Raman & Hassanaly, 2019). To this end, recent work has observed the effect of smoothed latent trajectories for advection-dominated systems produced by reduced neural ODE simulations, pointing to a relationship between model acceleration and intrinsic time-scale elimination provided by the latent space (Wan et al., 2023). Similar trends have been shown for neural ODE based surrogate models for stiff chemical kinetics (Vijayarangan et al., 2023). Despite these advances, however, the source of acceleration in the overall modeling strategy remains unclear: the goal of achieving model accuracy from both forecasting and autoencoding perspectives often overshadows the need to identify the contribution of each of these components to the empirically observed model acceleration (i.e., the need to interpret the dynamical properties of the learned neural ODE). Quantitative insight into the sources of acceleration – particularly, the degree of time-scale elimination – provided by the neural ODE based ROM can lead to valuable physical and model-oriented insights, and is the scope of the current study.

To understand and interpret sources of neural ODE acceleration, a more rigorous and quantitative analysis of time-scales produced by the neural ODE based ROM is warranted. This is accomplished here through a direct study of the effect of critical neural ODE training parameters – such as the overall integration time (training trajectory length) used to optimize models – on the accuracy and degree of time-scale elimination produced in the latent space. To execute this analysis, neural ODE Jacobians are used to directly quantify the fastest and slowest time-scales in the learned autoencoder-generated latent space. Ultimately, this study demonstrates the critical role of training trajectory length as a controlling parameter for time-scale elimination and model acceleration in latent spaces of advection-dominated systems. Demonstration studies are performed using ground-truth data from numerical simulations of the Kuramoto-Sivashinsky (KS) equation, and further tests are conducted for a hydrogen-air channel detonation configuration (Appendix A). The study concludes by demonstrating how these latent time-scales relate to the underlying physics of the full-order system. Specifically, for the KS equation, the largest (slowest) time-scales of the best performing neural ODE are found to recover the largest time-scales of the full-order system.

## 2   METHODOLOGY

**Dataset:** The full-order system dataset comes from numerical solutions of the KS partial differential equation. The KS equation is solved using `jax-cfd` (Kochkov et al., 2021), which generates ground-truth solution trajectories on a uniform one-dimensional spatial domain using a pseudospectral discretization. [1] Time integration is performed with an explicit Euler method. Training data is generated from 25 distinct trajectories of 1500 instantaneous full-order solution snapshots separated by a fixed time-step of $\Delta t = 0.01$. Each snapshot is 512-dimensional and resides in a one-dimensional physical space. During training, trajectories are broken into sub-trajectories of length specified by the training trajectory length $n_t$. Trained models are then tested using new initial conditions. The reader is referred to the recent work of Wan et al. (2023) for details of the KSE formulation and initial condition generation.

**Neural ODE for Latent Dynamics:** Ground-truth physical space trajectories are projected to a reduced latent space using an autoencoder. The neural ODE utilizes these latent trajectories to learn a continuous-time model of the latent dynamical system given by

$$\frac{d\widetilde{\mathbf{w}}(t)}{dt} = \mathcal{N}(\widetilde{\mathbf{w}}(t)), \quad \widetilde{\mathbf{w}}(t_0) = \mathbf{w}(t_0), \tag{1}$$

where $\mathcal{N}$ is a neural network and $\widetilde{\mathbf{w}}(t)$ represents the predicted latent variable. The parameters of the neural network are trained by minimizing the objective function

$$\mathcal{L}_{\text{NODE}} = \left\langle \frac{1}{n_t} \sum_{j=1}^{n_t} \left\| \mathbf{w}(t_0 + j\Delta t) - \widetilde{\mathbf{w}}(t_0 + j\Delta t) \right\|_2^2 \right\rangle, \tag{2}$$

where the target latent variable $\mathbf{w}(t) \in \mathbb{R}^{N_w}$ is obtained by applying an encoder $\phi$ to the corresponding physical space snapshot $\mathbf{u}(t) \in \mathbb{R}^{N_u}$, such that $\mathbf{w}(t) = \phi(\mathbf{u}(t))$ and $N_w \ll N_u$. The neural

---

[1]In this work, the domain length is 64, and is discretized into 512 points. The KS viscosity is set to 0.01.

ODE formulation in Eq. 2 is advantageous as it is dynamics-informed, training the latent dynamical system to minimize error across all $n_t$ time steps. As such, the training trajectory length, $n_t$, is a key parameter, selected *a-priori*, that influences the amount of dynamical information used to construct the latent model. Varying $n_t$ values in training is therefore expected to lead to neural ODEs with differing predictive and stability characteristics based on the intrinsic time-scales of the full-order system.

**Latent Time-Scales:** Jacobian eigenvalue analysis is incorporated here to extract dynamical system time-scales (Maas & Pope, 1992; Mease et al., 2003; Valorani & Goussis, 2001). Specifically, this analysis is employed to evaluate the impact of $n_t$ and neural network architecture hyperparameters on latent time-scales. Given Eq. 1, the fastest (limiting) and slowest time-scales in the latent space can be computed as

$$t_{\text{lim}}(t) = \frac{1}{\max(|\text{eig}(\frac{\partial \mathcal{N}(\widetilde{\mathbf{w}}(t);\theta_{\mathcal{N}})}{\partial \widetilde{\mathbf{w}}})|)} \quad \text{and} \quad t_{\text{max}}(t) = \frac{1}{\min(|\text{eig}(\frac{\partial \mathcal{N}(\widetilde{\mathbf{w}}(t);\theta_{\mathcal{N}})}{\partial \widetilde{\mathbf{w}}})|)}, \quad (3)$$

respectively. $t_{\text{lim}}$ is indicative of the largest feasible explicit time-step for evolution in the latent space, and $t_{\text{max}}$ is a measure of the slowest-evolving dynamical features (as discussed below, this work aims to correlate $t_{max}$ with neural ODE predictive accuracy). Additionally, since `jax-cfd` is used to solve the KS equation, differentiation through the full-order (physical space) dynamical system $d\mathbf{u}/dt = \mathbf{F}(\mathbf{u})$ is also possible, which facilitates Jacobian computation $\frac{\partial \mathbf{F}}{\partial \mathbf{u}}$ for extracting the corresponding *true* physical space time-scales in the same manner.

**Architectures, Training, and Evaluation:** The neural ODE architecture comprises a feed-forward neural network with four hidden layers (120 neurons in each hidden layer) utilizing ELU activation functions. The encoder architecture features a series of 1D convolutional layers with batch normalization and ELU activation function, designed to gradually reduce the spatial dimension of the physical space input. This down-sampling process halves the spatial component while doubling the channel count, leading to a flatten operation and a linear layer that creates the latent space representation $\mathbf{w}(\mathbf{t})$. The decoder mirrors the encoder design, substituting convolution layers with transpose convolution layers. Architectures are implemented in `PyTorch` (Paszke et al., 2019), and `torchdiffeq` (Chen, 2018) is leveraged for neural ODE training routines. The ADAM optimizer was used for optimization with initial learning rate set to $10^{-3}$. Two distinct training methodologies to train the autoencoder and neural ODE were compared: a *decoupled* approach, where the autoencoder and neural ODE are trained separately, and a *coupled* approach, where the two are trained simultaneously in an end-to-end manner. For rollout predictions, the decoupled approach was found to give models with better predictive accuracy. To compare different models' predictive accuracy, mean squared ($\mathcal{L}_{\text{RO}}$) and relative absolute ($\mathcal{R}_{\text{RO}}$) rollout errors are used. $\mathcal{L}_{\text{RO}}$ is the same as Eq. 2, but with different rollout $n_t$ values. $\mathcal{R}_{\text{RO}}$ is similar, but measures the absolute difference between the ground truth and predicted values, normalized by the ground truth value, and is commonly used in the neural ODE literature.

## 3  RESULTS

Figure 1 (left) shows limiting time-scales $t_{\text{lim}}(t)$ in the latent space for autoencoders with different latent dimensions $N_w$, convolutional layer counts, and training trajectory lengths ($n_t$) under both coupled and decoupled training methods. It can be seen that the number of convolutional layers and latent space dimensionality do not have a significant impact on the limiting time-scales. As for the effect of the training trajectory length $n_t$, the figure demonstrates that increasing the training trajectory length increases the limiting time-scale in latent space for both decoupled and coupled training methods, resulting in smoother trajectories and larger permissible explicit time-steps. Specifically, in both coupled and decoupled training strategies, raising $n_t$ from 100 to 4000 markedly amplifies limiting time-scale in latent space by roughly two orders of magnitude. It is evident that among all the studied hyperparameters, the training trajectory length ($n_t$) is the sole parameter with a substantial impact on $t_{\text{lim}}$.

Figure 1(e) and (f) compare, respectively, the ground truth spatio-temporal evolution of the KSE velocity field $\mathbf{u}$ with rollout predictions by the autoencoder neural ODE framework, which was trained using $n_t = 500$ in a decoupled manner. Corresponding target ($\mathbf{w}$) and predicted ($\widetilde{\mathbf{w}}$) latent space trajectories for $n_t = 500$ and 4000 are provided in Fig. 1(g). It can be seen that the latent space

trajectories predicted by the neural ODE with $n_t = 500$ more closely matches the ground truth trajectories as compared to the neural ODE with $n_t = 4000$. These noticeable differences in accuracy suggest that the relationship between increasing $n_t$ and predictive strength is not straightforward, as discussed later. Additionally, latent space trajectories are smoother than state space fields, with $n_t = 4000$ trajectories being smoother than $n_t = 500$, reflecting an increase in $t_{lim}$ with larger $n_t$.

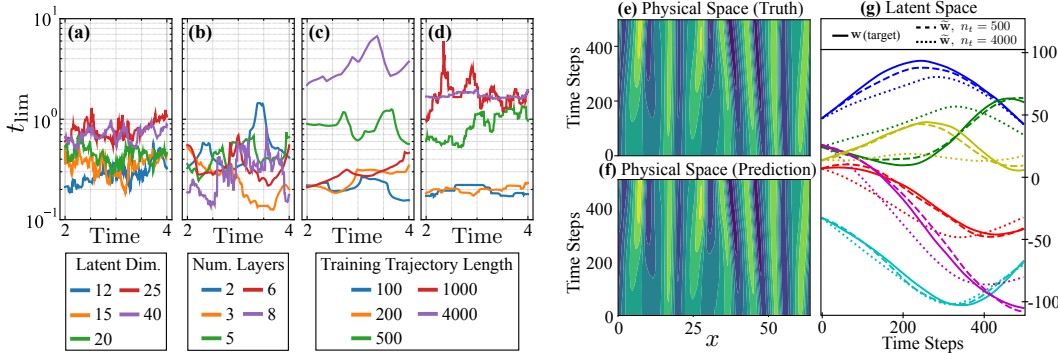

Figure 1: **(Left)** Limiting time-scale in the latent space $(t_{lim}(t))$ for neural ODEs trained with varying: **(a)** number of latent dimensions (with $n_t = 500$ and 4 encoder convolution layers), **(b)** number of autoencoder convolutional layers (with $n_t = 500$ and latent dimension of 25), **(c)** $n_t$ using a decoupled training approach, and **(d)** $n_t$ using a coupled training approach. Latent dimension of 25 and 4 convolution layers are used in (c) and (d). **(Right)** Neural ODE (latent dimension of 25 and 4 convolution layers) predictions for an unseen initial condition, where **(e)** shows ground-truth trajectory in physical space, **(f)** shows decoded rollout predictions in physical space for an $n_t = 500$ model using a decoupled training approach, and **(g)** shows corresponding latent space trajectories (solid line is target latent variable $\mathbf{w}$, dashed is predicted $\widetilde{\mathbf{w}}$ for $n_t = 500$, and dotted is predicted $\widetilde{\mathbf{w}}$ for $n_t = 4000$).

Figure 2 displays the mean-squared ($\mathcal{L}_{\text{RO}}$) and relative absolute ($\mathcal{R}_{\text{RO}}$) errors for neural ODEs trained using varying $n_t$ values in a decoupled training approach, across a 500 time-step predicted rollout trajectory. It can be seen that the rollout losses initially rise with increasing $n_t$, and reach a minimum at an optimal $n_t$ of 500, indicating this value best represents the system's physics for enhanced predictive accuracy at this tested rollout length. Although this may seem expected, as the optimal $n_t$ coincides with the trajectory length used to evaluate the error, identification of this 'optimal' $n_t$ is crucial for achieving maximum prediction precision. Figure 2 also shows $t_{\max}(t)$ for neural ODEs with different training trajectory lengths $n_t$. Interestingly, neural ODEs with $n_t = 8$ and $n_t = 4000$ have $t_{\max}$ values diverging from the full system, while $n_t = 500$ aligns closely, correlating with the lowest rollout errors. This suggests neural ODEs with the best accuracy have $t_{\max}$ values matching the full system's slowest physics. Although determining the exact optimal $n_t$ *a-priori* is still an open problem, analyzing $t_{max}$ can guide the selection of effective $n_t$ values for maximizing predictive accuracy.

## 4 Conclusion

This study explores the impact of training parameters on latent time-scales within neural ODE-based surrogate models. Key insights include the pivotal role of training trajectory length in influencing the latent system time-scales and model acceleration. The eigenvalue analysis method employed here proves effective in quantifying the fastest and slowest time-scales in the latent space, further highlighting the critical importance of training trajectory length ($n_t$). This study also underscores the significance of the optimal $n_t$ in neural ODE-based models, emphasizing its correlation with the largest eigenvalues of the system's dynamics. The optimal $n_t$ is crucial for achieving predictive accuracy, as it aligns the neural ODE's time-scales with the full system's slowest evolving physics. This correlation between $n_t$ and the largest eigenvalues offers a novel perspective in neural ODE training, guiding future research and practical applications in dynamical system modeling.

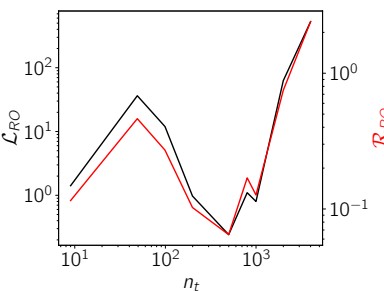 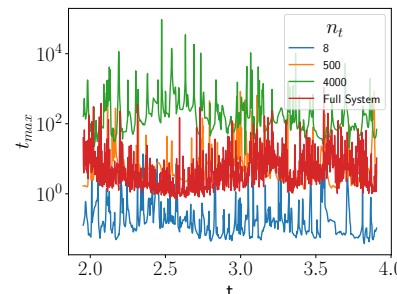

Figure 2: (Left) Mean squared ($\mathcal{L}_{\text{RO}}$) and relative absolute ($\mathcal{R}_{\text{RO}}$) rollout errors as functions of $n_t$ (Right) Largest time-scale in the latent space $t_{max}(t)$ for neural ODEs with different training trajectory lengths $n_t$.

ACKNOWLEDGMENTS

The manuscript has been created by UChicago Argonne, LLC, Operator of Argonne National Laboratory (Argonne). Argonne, a U.S. Department of Energy Office of Science laboratory, is operated under Contract No. DEAC02-06CH11357. The U.S. Government retains for itself, and others acting on its behalf, a paid-up nonexclusive, irrevocable worldwide license in said article to reproduce, prepare derivative works, distribute copies to the public, and perform publicly and display publicly, by or on behalf of the Government. This research used resources of the Argonne Leadership Computing Facility, which is a U.S. Department of Energy Office of Science User Facility operated under contract DE-AC02-06CH11357. SB and PP acknowledge funding support from U.S. Department of Energy (DOE) through the Argonne Energy Technology and security (AETS) fellowship under contract DE-AC02-06CH11357. RM acknowledges funding support from DOE Advanced Scientific Computing Research (ASCR) program through DOE-FOA-2493 project titled "Data-intensive scientific machine learning".

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

## A EXTENSION TO UNSTEADY DETONATIONS

To better evaluate how the time-scale trends in the autoencoder-neural ODE framework generalize beyond the KS equation, this section serves as an extension of the approach to unsteady detonations, a more realistic and highly advection-dominated benchmark problem. In broad terms, detonations are being actively explored as driving modes for combustion in next-generation power and propulsion concepts (Raman et al., 2023; Shepherd, 2009). The dynamical complexity of the detonation wave front motivates development of surrogate models that accelerate the solution of respective governing equations, which are the compressible reacting Navier-Stokes equations. Considering the analysis of the KS equations in the main text, the goal of this section is to highlight the consistent relationship between training trajectory length ($n_t$) and limiting latent timescale ($t_{\mathrm{lim}}$) across fundamentally different PDEs.

Training data is generated using resolved simulations of the compressible reacting Navier-Stokes equations in one spatial dimension in a standard channel detonation configuration. The resulting dataset can be described by the unsteady, self-sustained propagation of a hydrogen-air detonation wave through the spatial domain, where a detonation is characterized as a shock-wave coupled with a strong chemical reaction zone (see Fig. 3). Data is generated using a solver based on UMReactingFlow (Bielawski et al., 2023) built on the AMReX framework (Zhang et al., 2019), with chemistry modeled using the detailed chemical kinetic mechanism from Mueller et al. (1999). For a complete description of the simulation configuration and dataset, the reader is directed to Nair et al. (2024). In summary, two detonation trajectories are simulated to populate the dataset, each described by the pressure ratio between the driver and ambient gas, $P_{\mathrm{d}}/P_{\mathrm{amb}}$, with testing set evaluations performed on the same pressure ratios, but extrapolated in time. Each temporal snapshot captures fluid density, pressure, temperature, velocity, and species mass fractions at each spatial point (6000 total spatial points), giving an input channel depth of 13, unlike the unity depth in the KS equation. The snapshots were recorded at intervals of $\Delta t = 10^{-8}$s.

The predictive accuracy of the autoencoder-based neural ODE, trained with a decoupled approach using $n_t = 250$, latent dimension of 10, and 4 encoder convolutional layers, is assessed by extrapolating beyond the training data until the detonation exits the domain. Figure 3 shows predicted and target profiles for pressure, temperature, and mass fractions of two intermediate species ($H_2O_2$ and $H_2O$), indicating that the model effectively extrapolates beyond the training set for different pressure ratios and can properly represent the self-sustained detonation propagation.

Figure 4 (left) shows the limiting time-scale, $t_{\mathrm{lim}}$, in the latent space versus time for neural ODEs trained with various $n_t$ values (using latent dimension of 10 and 4 encoder convolution layers). Although the configuration is significantly different here, Fig. 4 reveals that higher $n_t$ generally increases $t_{\mathrm{lim}}$ in a similar manner as the KS equation trends shown in Fig. 1. Although not shown here, the latent variable evolution – which captures the propagation of detonation wave front (a shock coupled with a cheimcal reaction zone) – was also found to exhibit smoothness for this problem, which has significant implications for acceleration of detonation-containing flow simulations. Interestingly, Fig. 4 (right) – which shows the rollout error of predicted trajectories (evaluated using 1250 rollout steps) as a function of $n_t$ – also indicates an optimal $n_t$ value akin to observations in Figure 2 for the KS equation. Although this optimal value is present, it should be noted that comparing the neural ODE limiting time-scales ($t_{\mathrm{lim}}(t)$) to their full-system counterparts in this configuration (i.e., producing an analog to Fig. 2 (right) for the detonation) was not possible, since this requires full-system Jacobians ($\frac{\partial F}{\partial u}$) of the compressible reacting Navier-Stokes equations and the solver used here was not differentiable. However, the general consistency in trends with respect to the training trajectory length is promising, and motivates further neural ODE analysis for complex advection-dominated configurations in future work.

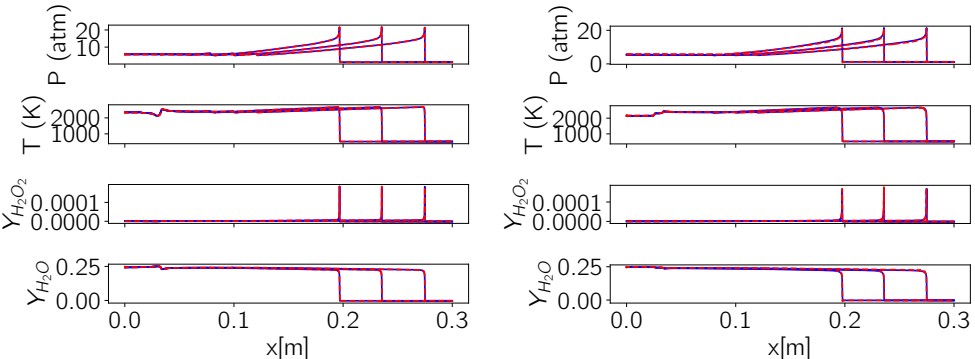

Figure 3: Comparison of the ground truth (solid blue) and predicted (dashed red) fields for outside the training set for (left) $P_\mathrm{d}/P_\mathrm{amb} = 40$ and (right) $P_\mathrm{d}/P_\mathrm{amb} = 20$. Results are shown for $n_t = 250$, latent dimension of 10, and four convolutional layers, with extrapolation time of 37.5 $\mu s$ (7500 time-steps).

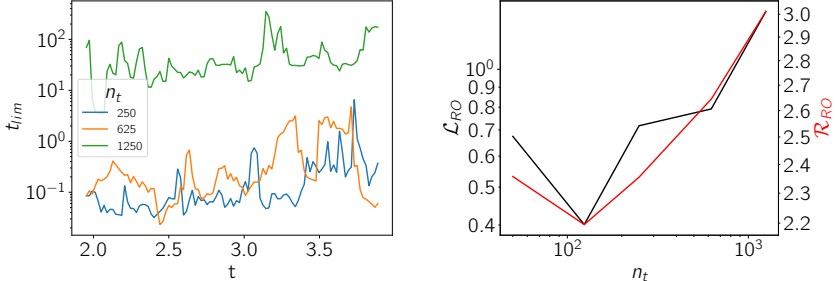

Figure 4: (Left) Limiting time-scale in the latent space $t_\mathrm{lim}$ for neural ODEs with different training trajectory lengths $n_t$. (Right) Mean squared rollout error $\mathcal{L}_\mathrm{RO}$ and relative rollout error $\mathcal{R}_\mathrm{RO}$ as functions of $n_t$ for a rollout trajectory length of 1250 time-steps.

