# OpenReview forum: "Investigation of Latent Time-Scales in Neural ODE Surrogate Models"
_ICLR.cc/2024/Workshop/AI4DiffEqtnsInSci — AI4DiffEqtnsInSci @ ICLR 2024 Poster_

### Official Review · Reviewer_whdy · 2024-02-20
**Interesting but limited experiments**

**Rating:** 5
**Confidence:** 3

**Review:**

### Summary
This work studies the relation between the (fastest/slowest) time-scales of a dynamical system and the time scales of an autoencoder-based neural ODE surrogate in its latent space. The authors show empirically for the Kuramoto-Sivashinsky equation that the autoencoder architecture (number of layers and latent dimension) does not impact significantly the time scales of the latent space, while the length of the training trajectories significantly correlates with the slowest/fastest time.

### Comments
* The subject is interesting and well suited to the workshop.
* The goal of the paper is clearly presented and well introduced, although a bit verbose.
* The results are interesting but very limited and mainly empirical. In particular a single dynamical system is used for the experiments. It is hard to know if the results carry for other systems. The architecture (and training) of the neural ODE could also have an impact, which is not considered.

---

### Official Review · Reviewer_gkug · 2024-02-26
**The paper is well-written and structured but its generalizability remains limited.**

**Rating:** 6
**Confidence:** 3

**Review:**

The paper explores the impact of training parameters on latent time-scales within a neural ODE-based surrogate model of the Kuramoto-Sivashinsky equation. It also investigates the significance of the trajectory length and relates it to the largest eigenvalues of the system.

Pros
- The paper is well-structured, the plots are well-documented and good to understand.
- The focus on trajectory lengths makes it easy to follow along. The results show the sensitivity of the accuracy of results to trajectory length.

Quality: The quality of the paper is good. \
Clarity: The paper is well-structured. \
Originality: good. \
Significance of this work: The results presented are promising. However, to determine the significance and expected generalizability--given the title and abstract--a more elaborate benchmark would be needed.


Major issues
- Generalizability: As only one ODE is investigated, the generalizability of results is unclear.
- One of the main results of the authors is the correlation between training length and the largest eigenvalue of the system dynamics/t_lim. This is also visualized and described in Figure 1c/d. However, it is not clear how exactly they are correlated. For example: t_lim is very similar for decoupled approach with trajectory length 1000, 100 and 200 but higher for 500, i.e. there is neither a clear positive nor negative correlation. An investigation of average t_lim values over time (x-axis) and more training trajectory lengths could help investigate this correlation better.


Minor issues
- The paper title might be misleading. The authors only investigate the KS equation. In the abstract (+introduction) the authors specify that the subject of the paper are "surrogate models for advection-dominated dynamical systems". Generalizability is unclear.
- Why did the authors choose n_t= 500 and n_t=4000 for Figure 1g?
- Is it n_t=7 oder n_t=8 in Figure 2 (right)?
- Acceleration is not mentioned in the discussion.

---

### Meta-Review · Area_Chair_DZay · 2024-02-26

**Recommendation:** Accept (Poster)

**Metareview:**

The reviewers both agree that the topic is a good fit for this workshop. Both mention that the results are only empirical and limited by showing only 1 ODE. For a workshop paper,  I think that is ok as preliminary work. I vote for acceptance and recommend for a full submission that the authors test their method on more systems.

---

### Decision · Program_Chairs · 2024-02-28

Accept (Poster)